# Engineering Nanofiber Scaffolds with Biomimetic Cues for Differentiation of Skin-Derived Neural Crest-like Stem Cells to Schwann Cells

**DOI:** 10.3390/ijms231810834

**Published:** 2022-09-16

**Authors:** Ashis Kumar Podder, Mohamed Alaa Mohamed, Georgios Tseropoulos, Bita Nasiri, Stelios T. Andreadis

**Affiliations:** 1Department of Chemical and Biological Engineering, University at Buffalo, The State University of New York (SUNY), Buffalo, NY 14260, USA; 2School of Pharmacy, Brac University, Dhaka 1212, Bangladesh; 3Department of Chemistry, Faculty of Science, Mansoura University, Mansoura 35516, Egypt; 4Department of Biomedical Engineering, University at Buffalo, The State University of New York (SUNY); Buffalo, NY 14260, USA; 5Center of Excellence in Bioinformatics and Life Sciences, University at Buffalo, The State University of New York (SUNY), Buffalo, NY 14260, USA; 6Center of Cell, Gene and Tissue Engineering (CGTE), University at Buffalo, The State University of New York (SUNY), Buffalo, NY 14260, USA

**Keywords:** neural crest, Schwann cell, nanofiber, NRG1, poly-ε-caprolactone, topographical cues, keratinocyte derived neural crest, proteolipid protein PLP1, S100b

## Abstract

Our laboratory reported the derivation of neural crest stem cell (NCSC)-like cells from the interfollicular epidermis of the neonatal and adult epidermis. These keratinocyte (KC)-derived Neural Crest (NC)-like cells (KC-NC) could differentiate into functional neurons, Schwann cells (SC), melanocytes, and smooth muscle cells in vitro. Most notably, KC-NC migrated along stereotypical pathways and gave rise to multiple NC derivatives upon transplantation into chicken embryos, corroborating their NC phenotype. Here, we present an innovative design concept for developing anisotropically aligned scaffolds with chemically immobilized biological cues to promote differentiation of the KC-NC towards the SC. Specifically, we designed electrospun nanofibers and examined the effect of bioactive cues in guiding KC-NC differentiation into SC. KC-NC attached to nanofibers and adopted a spindle-like morphology, similar to the native extracellular matrix (ECM) microarchitecture of the peripheral nerves. Immobilization of biological cues, especially Neuregulin1 (NRG1) promoted the differentiation of KC-NC into the SC lineage. This study suggests that poly-ε-caprolactone (PCL) nanofibers decorated with topographical and cell-instructive cues may be a potential platform for enhancing KC-NC differentiation toward SC.

## 1. Introduction

Development of scaffolds that mimic the architecture, physical characteristics, and biological functions of the native ECM has received a lot of attention due to their potential for tissue regeneration [1,2]. The structural anisotropy and fibrous architecture of ECM, with fiber diameters spanning from nanometer to micrometer scale, have inspired the development of fibrous scaffolds [3], prepared by phase separation [4,5], self-assembly [6,7,8], electrospinning [9,10,11,12] and microfluidic approaches [13,14,15]. Among these, electrospinning has attracted considerable attention because it can provide fibers with a high surface-to-volume ratio, pore-connectivity, and ease of tuning fiber diameter and porosity by controlling simple physical parameters such as applied voltage, feed-rate, and tip-to-collector distance. Electrospinning can employ synthetic or natural polymers [16] of a wide range of concentrations [17], viscosities [18], molecular weights, and solvents [16,19]. Synthetic materials exhibit good and easily controlled mechanical properties but often lack the bioactivity required to trigger a specific cell response. On the other hand, biopolymers contain bioactive cues required for cell attachment, spreading, and proliferation but suffer from a lack of mechanical strength [20]. We hypothesized that fibrous scaffolds prepared by a combination of synthetic and natural polymers may exhibit mechanical properties and bioactivity that mimic the native ECM and promote stem cell differentiation to Schwann cells.

Fibrous scaffolds made from polyesters such as PCL, polylactic acid (PLA), polyglycolic acid (PGA), and their copolymers have been explored mainly to direct the fate of the neuronal and glial cells [21]. Post-synthesis modification strategies such as crosslinking chemistries have been used to modify the mechanical properties [22,23,24,25,26,27,28,29,30], while others such as plasma treatment [31], polydopamine (PDA)-coating [32], hydrolysis [33], aminolysis [34], and layer-by-layer coating [35] have been employed to introduce chemical functionality to the fiber surface. However, immobilization of biomolecules is limited by the chemical functionality, and therefore, development of novel approaches that enable surface functionalization with a broad range of bioactive signals is critical to directing stem cell fate decisions.

In vertebrates, a unique stem cell population of NC cells arises and migrates along the neural tube during the gastrulation phase of embryo development [36]. Following the formation of the neural tube, these cells migrate out to the distal parts of the embryo and give rise to different types of cells of the peripheral nervous system (PNS), such as peripheral neurons and SCs. Additionally, they differentiate towards chondrocytes and osteoblasts of the craniofacial skeleton, melanocytes, adipocytes, and smooth muscle cells [37]. Among them, SCs are responsible for myelinating the neuronal axons of the PNS and maintaining normal nerve function by facilitating electrical current propagation through the nodes of Ranvier along the myelin sheath. SCs have a remarkable ability to regenerate the PNS by removing the myelin debris from the injury site and differentiating into myelinating SCs [38,39]. However, this is a long and often inefficient process that may require external intervention such as surgical nerve transfer and adjunct SC therapy to achieve satisfactory functional recovery [40]. Moreover, it is estimated that about 6 out of 100 Americans suffer from peripheral nerve injury (PNI) caused by trauma or medical conditions, such as diabetes and tumor-related surgeries [41,42]. Thus, there is a dire need for efficient and scalable approaches to obtain large numbers of functional SC for PNI cell therapy.

Nerve biopsies are one of the sources of primary SC but are limited by donor availability and the cumbersome ex-vivo expansion of the isolated cells [43]. Another source is human fetal tissue, but it is plagued by ethical concerns, small number of isolated cells, and lack of scalability [36]. Induced pluripotent stem cells (iPSC) are a viable alternative source of NCSC [44,45,46,47,48], but induction of iPSC requires genetic modification to express essential pluripotency genes, which may result in tumorigenesis. Others have attempted to directly differentiate adult human cells such as fibroblasts into SC via inhibition of TGF-β, BMP, and GSK3 signaling [49]. More recently, NCSC-like cells were derived from the bulge of the hair follicle [42], and our lab reported the derivation of NCSC-like cells from the interfollicular epidermis of neonatal foreskin [50], as well as the skin of older adults [51], by activating the PI3K/Akt and MAPK/Erk1/2, and suppressing the TGFβ pathway [50,52]. KC-NCs could differentiate into functional neurons, SCs, melanocytes, and smooth muscle cells in vitro. Most notably, KC-NCs migrated along stereotypical pathways and gave rise to multiple NC derivatives upon transplantation into chicken embryos [50]. However, current protocols for guiding the differentiation of NCSC into SC have room for significant improvement [42,53]. Here we provide a facile approach for developing nanofibers with topographical and cell-fate determining cues to enhance the differentiation of KC-NC into SC.

## 2. Results

### 2.1. Fiber Formation and Decoration with Biological Cues

We developed an innovative fibrous scaffold to provide topographical and biological cues to guide the differentiation of KC-NC towards Schwann cell fate. The design strategy was based on the development of aligned poly-ε-caprolactone (PCL)-based nanofibers using electrospinning (Figure 1A); followed by coating the fiber surface with polydopamine (PDA) (Figure 1B) (2 mg/mL dopamine solution in 10 mM Tris–HCl buffer, pH = 8.5) [32]. After washing thoroughly with DI water to remove any loosely bound PDA, the fibers were chemically conjugated with Hep-thiol (Figure 1C) via a Michael-addition reaction as described below. This allows the immobilization of bioactive molecules containing heparin-binding domain (HBD) including growth factors such as PDGF-BB, FGF2, NRG1 (Figure 1D) [54], or recombinant molecules containing HBD. For the immobilization of native ECM proteins such as laminin and fibronectin, the PDA-modified fibers were submerged into the respective solutions and coated overnight. This design allows for flexibility in the immobilization of biomolecules for manipulation of cell alignment and differentiation (Figure 1E).

PCL-PDA was functionalized with heparin to enable the immobilization of biomolecules through their HBD. First, Hep-thiol was synthesized by an amidation reaction between the carboxylic group of heparin and DTPDH in the presence of the EDC coupling agent at pH 4.75, followed by reductive cleavage of disulfide bonds using TCEP (Appendix A). ^1^H NMR analysis confirmed the chemical structure of Hep-thiol, and characteristic signals from C*H*_2_–SH and C*H*_2_–CH_2_–SH appeared at 2.7 and 2.8 ppm, respectively (Figure 2A). Second, Hep-thiol was attached chemically to PCL-PDA via Michael-Addition chemistry between SH groups and the PDA ring (Figure 2B). Oxidative polymerization of dopamine on PCL fibers at pH 8.5 resulted in PCL-PDA fibers.

Successful coating with PDA was observed visually by the color change of PCL fibers from white (unmodified) to deep brown (coated with PDA) (Figure 2C) and the chemical structure of PCL, PCL-PDA, and PCL-PDA-Hep fibers were confirmed by FTIR analysis (Figure 2D and Appendix A). The characteristic symmetric and asymmetric stretching vibrations of CH_2_ groups from PCL appeared at 2866 and 2944 cm^−1^, respectively, while the distinct peak of C=O and C-O-C bonds appeared at 1722 and 1167 cm^−1^, respectively. The FTIR spectrum of PCL-PDA showed the characteristic signal from catechol deformation at 1610 cm^−1^ (Figure 2D, orange band). Finally, a broad peak between 3066–3600 cm^−1^ was ascribed to the stretching vibration of -COOH and -OH groups from heparin (Figure 2D, green band).

The surface morphology of the electrospun fibers was investigated using SEM. The electrospun PCL exhibited a fibrous morphology with a well-aligned structure and smooth surface (Figure 3A(i,iv)). The average fiber diameter was measured from the SEM images and was found to be 823 nm **(**Figure 3B), which is close to the size of the neuronal axon diameter of ~1 μm. The orientation angle of fibers revealed a slight deviation from the right angle for most fibers with an average orientation angle of 89.8 ± 11.8° (Appendix A). The electrospun PCL showed 81.4% of the fibers between 0–10° deviations and 14.2% between 10–20°, confirming successful alignment (Appendix A). The SEM images of PCL-PDA and PCL-PDA-Hep revealed that fiber alignment was preserved after coating with PDA and chemical attachment of Hep-thiol (Figure 3A(ii,v) and (iii,vi)), respectively. Specifically, the orientation angle for PCL-PDA-Hep was measured to be 89.7 ± 9.2° (Figure 3C), with 77.4% of the fibers deviating by 0–10°, and 19.2% by 10–20°, confirming that alignment (Appendix A). Finally, PCL-PDA and PCL-PDA-Hep exhibited rough fiber surfaces compared with PCL due to the formation of nanoparticles from PDA coating.

The bulk mechanical properties of the modified fibers were measured by tensile test. The stress-strain curve of the PCL-PDA-Hep showed a steep increase in the stress until the break, revealing high strength (Figure 3D). Young’s modulus, elongation at break, and ultimate tensile strength were calculated from the stress-strain curve and found to be 18.9 MPa ± 1.0, 19.1% ± 0.8, and 2.7 ± 0.2 MPa, respectively.

### 2.2. Effect of Biomolecules on Adhesion and Spreading of KC-NC

Next, we tested the ability of KC-NC to attach and spread on the fibers that were decorated with native ECM or recombinant proteins containing HBD. All the PCL fibers were modified with PDA, followed by UV sterilization and coating with native ECM glycoproteins such as laminin and fibronectin. To immobilize proteins containing HBD, the fibers were decorated with heparin (2 mg/mL Hep-thiol solution at pH 7.4, PCL-PDA-Hep), followed by overnight conjugation with bidomain heparin-binding peptides (HBD-REDV or HBD-RGD) or heparin-binding growth factors (PDGF-BB, FGF2 or NRG1). After five days, the cells were fixed and stained with phalloidin to visualize the microfilaments of their cytoskeleton.

Irrespective of the adhesion cues, KC-NCs were able to attach to and align in the direction of the nanofibers. With regards to initial attachment, the cells showed the highest adhesion to HBD-REDV and lowest onto PDGF-BB, while the number of attached cells was similar on laminin, fibronectin, HBD-RGD, FGF2, and NRG1 proteins (Figure 4I). KC-NC showed increased spreading on the fibers coated with fibronectin, HBD-REDV, FGF2, and NRG1, compared to the control fibers (*p* < 0.0001) (Figure 4J). The degree of cell spreading/elongation was determined by calculating the eccentricity (φ) of individual cells. When the value of φ approaches 1.0, it corresponds to a perfectly bipolar shape resembling a line segment, while a value closer to 0 indicates a circular shape (Figure 4K). Our results indicate low eccentricity in control and PDGF-BB coated fibers (*p* > 0.05). However, we observed a significantly higher degree of elongation of KC-NC along the fibers coated with NRG1, FGF2, fibronectin, HBD-REDV, and HBD-RGD (Figure 4J). This data suggests that specific ECM cues (fibronectin, HBD-REDV, HBD-RGD) and growth factors (FGF2, NRG1) elicit cytoskeletal changes, that enhance the spreading on PCL-PDA fibers. Among those, NRG1 is a key regulator of NC differentiation to SC, thereby providing a promising candidate for developing a platform for SC attachment and differentiation.

### 2.3. Effect of the Biological Cues on Differentiation of KC-NC Cells to SC Phenotype

Next, we evaluated the differentiation potential of the KC-NCs to SC-like cells using two markers of SC lineage, PLP1, and GFAP. In this experiment, KC-NCs were plated onto PCL-PDA-Hep fibers coated with Fibronectin, HBD-REDV, or NRG1 and differentiated towards SC cells for 14 days.

Immunostaining of the cells for both myelin PLP1 and GFAP indicated the presence of SC-like glial cells on all three substrates (Figure 5A–H). Quantification of the GFAP and myelin PLP1 fluorescence showed increased signal intensity per cell for the PCL fibers with fibronectin, HBD-REDV, and NRG1 as compared to control, uncoated nanofibers (Figure 5I,J). Moreover, the GFAP expression of cells on NRG1 immobilized fibers was significantly higher than those on the fibronectin or HBD-REDV groups (Figure 5I). On the other hand, PLP1 expression was significantly higher on all the substrates as compared to control fibers, with fibronectin showing lower levels than HBD-REDV and NRG1 (Figure 5J). These results suggest that NRG1 might be optimal among the immobilized proteins tested, as it promotes attachment, alignment, and differentiation of KC-NC into the Schwann lineage.

Indeed, in addition to PLP1 and GFAP, KC-NCs differentiated on PCL-PDA-Hep-NRG1 fibers showed significantly higher expression of two other markers, S100b and p75 (NGFR), as compared to control, PCL-PDA-Hep fibers (Figure 6A–J). Interestingly, cells on PCL-PDA-Hep-NRG1 fibers also showed higher levels of S100b and PLP1 but similar levels of the glial cell marker, GFAP, and pre-SC marker p75 NGFR as compared to the control tissue culture plates (TCP). Furthermore, cells seeded on the fibers showed reduced proliferation as compared to those on TCP (Figure 6K). Since cell differentiation and maturation are accompanied by decreased proliferation, NRG1-decorated nanofibers may provide a more physiological environment for NC differentiation toward SCs than traditional TCP.

Finally, we assessed whether immobilized NRG1 remained on the fiber surface, we measured the amount of NRG1 in the media (1% BSA in PBS) at various time points after immobilization using ELISA (Appendix A). The kinetics of NRG1 release in the media was determined over a period of 120 h. While NRG1 could bind to PCL-PDA fibers without heparin, almost 50% was released within 3 h. However, in the presence of heparin, only 1% of NRG1 was released in the first 3 h, with negligible release at later times (6–120 h), demonstrating almost irreversible binding of NRG1 on heparin for the time period tested.

## 3. Discussion

Aligned fiber scaffolds have been reported to support the unidirectional alignment of SC and elongation along the fiber axis, mimicking the developmental process of radial sorting, where SC precursors differentiate and envelop a single neuronal axon to form the myelin sheath and ensure efficient, long-range nerve signal conduction [55]. Primary SC (isolated from human fetal tissues) cultured on aligned fibers showed upregulation of the early myelination markers (MAG, MPZ) and downregulation of immature SC marker (NCAM) compared to cells on randomly spaced fibers or TCP [56]. In this study, we present a generalizable platform for SC maturation on PCL-based nanofibers with excellent alignment and surface chemistry that enables the immobilization of a broad range of biomolecules, including heparin-binding growth factors and bidomain peptides. When naive KC-NCs were seeded on these nanofibers, they attached and adopted a spindle-like morphology, similar to the native ECM microarchitecture of peripheral nerves. In addition, immobilization of biological cues promoted differentiation of KC-NCs into the SC lineage.

Previous studies reported protocols for differentiation of embryonic stem cells to SC [57] or hair follicle-derived NC to SC [42]. When we tested the first protocol with our KC-NC, we found that it worked well but differentiation took 5 weeks as evidenced by GFAP and S100b expression. The second protocol resulted in upregulation of SC markers within two weeks but after 17 days we observed extensive cell death, which was prohibitive for long-term experiments. In contrast, our differentiation protocol upregulated mature SC markers significantly, while keeping the cells alive for extended culture periods up to 5 weeks [50].

Previous studies attempted to recapitulate SC differentiation and maturation in vitro using fibrous substrates [58]. They identified fiber diameter as a key parameter regulating SC differentiation, with intermediate-size fibers showing the best results. Specifically, primary rat SCs exhibited the best alignment and faster migration on 1000 nm diameter fibers, as compared to larger fibers ranging from 5000 to 8000 nm in diameter [59]. Similarly, differentiation of human ESC-derived NCSC into SC progenitors was more efficient on aligned fibers with average diameters between 600–1600 nm as compared to fibers with smaller diameters (160 nm) or tissue culture plates. Interestingly, pre-differentiated NCSCs were more responsive to topological cues than undifferentiated ones [43]. Based on these results, we designed electrospun fibers with an average diameter of about 800 nm and examined the effect of bioactive cues in guiding KC-NC differentiation into SC differentiation on nanofibrous scaffolds.

Attachment of mussels to various types of surfaces occurs naturally due to the presence of 3,4-dihydroxy-L-phenylalanine (DOPA) on their surface [60]. Inspired by nature, we coated our nanofibers with PDA by polymerizing dopamine on the fiber surface, and further modified the quinone groups with Hep-thiol via the Michael addition reaction [61]. Immobilized heparin enabled the binding of heparin-binding growth factors such as PDGF-BB, FGF2, and NRG1, and two heparin-binding bidomain peptides, HBD-REDV and HBD-RGD, containing integrin binding peptides from fibronectin. PDGF-BB is a mitogen and survival factor, which is also known to affect NCSC migration [62], FGF2 is known to affect proliferation, and NRG1 guides differentiation of the NCSC and KC-NC into SC [42]. We also evaluated two native ECM glycoproteins, laminin, and fibronectin, which play a crucial role in governing NC migration during development and in response to peripheral neuropathy and subsequent nerve repair [63,64].

KC-NC exhibited elongated morphology on fibers functionalized with fibronectin and HBD-REDV in agreement with previous observations where fibronectin was reported to play a key role in the migration of NCs [65]. Indeed, integrin α_4_β_1_ is expressed on NC and binds to the fibronectin peptide REDV controlling avian NC cell migration, while inhibition of the α_4_β_1_ integrin was reported to reduce NC migration and apoptosis [66]. On the other hand, KC-NCs elongated less on laminin and HBD-RGD modified fibers, consistent with previous findings showing reduced numbers of migratory NC in neural folds upon exposure to laminin-α_5_ and increased migration in laminin-α_5_ mutant mice [67]. Among the growth factors, FGF2 and NRG1 promoted NC elongation compared to the PDGF-BB immobilized fibers. FGF2 is well known for regulating the proliferation of the NCSC and maintaining them in an undifferentiated state [68]. On the other hand, neuregulin1 (NRG1) is expressed on neuronal axons, where it promotes SC binding, axonal sorting, and myelination [69].

Compared to the PDA-immobilized fibers (control), biomolecule-functionalized fibers significantly enhanced KC-NC differentiation towards SC, as shown by expression levels of PLP1 and GFAP. While there was no significant difference in the expression of myelin PLP1 among HBD-REDV and NRG1, the NRG1-functionalized fibers significantly enhanced the expression of glial cell marker, GFAP, in agreement with a previous study that evaluated differentiation of ESC-derived NCSC on electrospun fibers [43]. NRG1 modified fibers also showed high expression of the pre-SC marker, p75 NGFR, and mature SC marker, S100b. Finally, we observed reduced cell proliferation on NRG1 nanofibers as compared to TCP, suggesting enhanced maturation of the KC-NC-derived SCs.

## 4. Materials and Methods

### 4.1. Materials

PCL (MW 50 kDa) was purchased from Polysciences (Warrington, PA, USA). Dopamine hydrochloride (98%), heparin sodium salt from porcine intestinal mucosa (180 USP units/mg), tris(2-carboxyethyl) phosphine (TCEP), and *N*-(3-dimethylaminopropyl)-*N*′-ethylcarbodiimide hydrochloride (EDC, 98%) were purchased from Sigma-Aldrich (St. Luis, MO, USA). 3,3’-Dithiobis(propanoic dihydrazide) (DTPDH) was obtained from Frontier Scientific (Logan, UT, USA). 1,1,1,3,3,3-Hexafluoro-2-propanol (HFIP) (99%) was purchased from Alfa Aesar (Ward Hill, MA, USA). All reagents were used as received unless otherwise noted.

#### 4.1.1. Development of Anisotropically Aligned PCL-PDA

Aligned PCL fibers were developed by electrospinning using a rotating drum as a collecting substrate. Specifically, PCL was dissolved in HFIP at a concentration of 20 wt%, and the electrospinning process was conducted at the following settings: applied voltage, 18 kV; feed rate, 1.5 mL/h; spinneret tip-to-collector distance, 12 cm; rotating drum speed, 1500 rpm. The electrospun mat was left to dry for one week at room temperature, followed by three days under vacuum. Afterward, the PCL fibers were immersed in a dopamine solution of 2 mg/mL (pH 8.5, Tris-HCl) at room temperature for 15 h to coat the PCL fibers with a layer of PDA. The PDA-coated PCL, named PCL-PDA, was washed thoroughly with deionized (DI) water and then used to conjugate the thiol group-containing heparin (Hep-thiol).

#### 4.1.2. Synthesis of Hep-Thiol

Heparin sodium salt (1.0 g, 8.3 × 10^−2^ mmol) and DTPDH (0.10 g, 0.42 mmol) were dissolved in DI water at a 5 mg/mL heparin concentration. The pH was adjusted to 4.75 using 1 M HCl. Afterward, EDC (8.0 × 10^−2^ g, 0.42 mmol) was added under stirring for 4 h. The pH was maintained at 4.75 during the reaction by adding 1 M HCl. Subsequently, TCEP (0.24 g, 8.4 mmol) was added, and the reaction mixture was stirred for an additional 3 h. Hep-thiol was purified by dialysis against the water of pH 3.5 for three days and then freeze-dried.

#### 4.1.3. Surface Functionalization with Hep-Thiol and Conjugation of the Biological Cues

PCL-PDA fibers were first sterilized by overnight incubation under UV light of a laminar flow hood and then mounted into a polystyrene chamber with a movable base (MatTek, Ashland, MA, USA). Hep-thiol solution at a concentration of 2 mg/mL (pH = 7–8, HEPES, 40 mM) was added to each well and incubated overnight (16 h). The fibers were washed three times with PBS to remove the unbound heparin and then stored with PBS at 4 °C till the next step. These fibers were termed as PCL-PDA-Hep. Before seeding the KC-NCs, PCL-PDA fibers were treated overnight with laminin (catalog #CC095, EMD Millipore, Chicago, IL, USA) and fibronectin (catalog# 33016015, Gibco, Thermo-Fisher, Philadelphia, PA, USA) at a concentration of 0.1 mg/mL. Recombinantly designed bi-domain fusion proteins (0.1 mg/mL) were incubated overnight with the PCL-PDA-Hep fibers to allow immobilization of the REDV and RGD peptides via their heparin-binding domain (HBD) domain. Heparin-binding growth factors such as PDGF-BB (Catalog #PHG0045, Gibco, Thermo-Fisher), FGF2 (Catalog #PHG0021, Gibco, Thermo-Fisher), and Heregulin/Neuregulin beta1/NRG1 (Catalog #SRP3055, Sigma, Atlanta, GA, USA ) were used at a concentration of 5 µg/mL in KC-NC induction media to coat the PCL-PDA-Hep fibers for at least 16 h.

#### 4.1.4. Fiber Characterization Measurements

The surface morphology of PCL fibers was determined using focused ion beam scanning electron microscopy (FIB-SEM, Carl Zeiss Auriga, Jena, Germany) at an applied voltage of 2.0 kV. To deposit a thin layer on the surface, the fibers were sputtered with gold for 40 s at 30 mA before SEM imaging. Fiber diameter and size distribution were measured from SEM images using Image J software (Version 1.53t, National Institute of Health, USA); a total of n = 44 fibers were analyzed. The orientation angle was measured from the SEM images using Image J software and the mean orientation angle was calculated by taking the average of n = 140 and n = 151 measurements for PCL and PCL-PDA-Hep, respectively. The percent frequency was calculated for angles at 5-degree intervals by dividing the number of fibers by the total number analyzed. Fourier transform infrared (FTIR) spectra of PCL, PCL-PDA, and PCL-PDA-Hep were recorded using an attenuated total reflection FTIR spectrometer (Vertex 70, Bruker, Billerica, MA, USA) in the range of 4000–400 cm^−1^ with a resolution of 4 cm^−1^. ^1^H NMR spectrum of Hep-thiol was obtained using a Varian INOVA-500 spectrometer at room temperature. The sample was dissolved in deuterium oxide (D_2_O) containing 1 vol % of tetramethylsilane (TMS) as the internal standard.

The mechanical properties of fibers were measured using Instron tensile tester model 3343 with 50 N load cell instrument (Instron, Norwood, MA, USA) at a crosshead speed rate of 100 mm/min to obtain a stress-strain curve. A rectangular specimen (20 mm length × 8 mm width × 0.13 mm thickness) was used for the test. All samples were measured in triplicate. Ultimate tensile strength and elongation at break were obtained directly from the stress–strain curve. Young’s modulus was calculated from the slope of the elastic region up to 10% strain on the stress (δ)–strain (ε) curves.

#### 4.1.5. Cloning and Production of the Recombinant Fusion Proteins

The core sequence of the two fusion proteins, Hep2-(GGGS-HIPREDVYH)_5_ and Hep2-(GGGS-GRGDS)_5,_ consist of two parts: (i) Hep2, the second heparin-binding domain (HBD) of fibronectin; (ii) R5, five tandem repeats, each composed of a flexible linker motif, GGGS followed by a peptide HIPREDVDYH or GRGDS, that are known for binding preferably to integrin α_4_β_1_ and α_v_β_1_, respectively. The Hep2 domain was cloned by RT-PCR from the second HBD of fibronectin using cloning primers (Table 1) containing the BamHI and XhoI cutting sites and inserted in the pET28a expressing vector. The R5 sequence of REDV containing cutting sites HindIII and XhoI was purchased from Invitrogen (Waltham, MA, USA) and cloned next to the H2 in the pET28a expression vector as described previously by Nasiri et.al. [70]. The cloning primers for the R5 sequence of RGD containing the cutting sites HindIII and XhoI were purchased from Invitrogen (Waltham, MA, USA) in two short fragments (Table 1).

Following the ligation of the two short fragments, they were cloned by inserting them next to the HBD domain of the pET28a expressing vector. The fusion proteins for both HBD-REDV and HBD-RGD were produced in the bacterial strain *Escherichia coli* BL21-DE3-pLysis. Specifically, bacteria were expanded until the optical density (OD) reaches a value of 0.7 and then induced with 0.1 mM isopropyl β-D-1-thiogalactopyranoside (IPTG) (Sigma-Aldrich. St. Luis, MO, USA) for protein production overnight at 22 °C and 240 rpm. The following day, the bacteria were centrifuged at 4000 rpm for 20 min, and the pellets were re-suspended in lysis buffer (500 mm NaCl (VWR Chemicals, LLC., Solon, OH, USA), in 1X PBS, pH 7.4), containing 1 mg/mL lysozyme (Sigma-Aldrich, St. Luis, MO), 1% Triton X-100 (Sigma-Aldrich. Inc., St. Luis, MO, USA) and 1 mM phenylmethanesulfonylfluoride (PMSF, Sigma-Aldrich. Inc., St. Luis, MO, USA) as protease inhibitor stirred for 1 h at RT followed by sonication for 10 cycles with 50% intensity, 30 s on/30 s off. The soluble protein was obtained by ultra-centrifugation of the sonicated lysate at 50,000× *g* for 15 min using the Avanti high-performance centrifuge (Beckman Coulter Inc., Indianapolis, IN, USA). Fusion proteins were then purified using HisTrap^TM^ HP Column (Cytvia, Uppsala, Sweden) following the manufacturer’s instructions. The purity of the collected protein was tested using 10% SDS-PAGE, where the fusion peptide was apparent at a molecular weight of ~40 kDa. Bradford assay was used to determine the final concentration of the proteins.

#### 4.1.6. Release Kinetics of the Immobilized NRG1 from PCL Fiber Surface

PCL fibers were functionalized with PDA and heparin as described previously and then NRG1 was immobilized on heparin. The fibers were treated with a solution of NRG1 (5000 ng/mL in 1% BSA in PBS) at 37 °C. The total medium was collected at the indicated time points as shown in Appendix A (3, 6, 12, 24, 48, and 120 h) and washed twice before replenishing with fresh 1% BSA/PBS solution. The samples were stored at −20 °C until use. The NRG1 (Sigma) concentration of each sample was determined by ELISA according to the manufacturer’s recommendations (Catalog# DY377, Duo set; R&D Systems, Minneapolis, MN, USA). A total of three independent experimental replicates were assessed for the NRG1 release kinetic study.

### 4.2. Cell Culture

#### 4.2.1. Epidermal Cell Isolation

Foreskin samples lacking hair follicles of 1 to 3-day-old neonates were obtained from the John R. Oishei Children’s Hospital, Buffalo. Skin tissues were washed three times with PBS, chopped into small pieces (~3 × 1 cm), and enzymatically digested with dispase II protease (Sigma, St. Louis, MO, USA) for 15–20 h at 4 °C. Afterward, the epidermis was separated from the dermis using fine forceps. The separated epidermis layer was further digested with Trypsin-EDTA (0.25%) (Life Technologies, Carlsbad, CA, USA) for 10–15 min at 37°C and filtered through a 70 μm cell strainer (BD Biosciences, Franklin Lakes, NJ, USA), centrifuged and plated on a confluent monolayer of growth-arrested 3T3/J2 mouse fibroblast feeder cells in keratinocyte growth medium (KCM) consisting of a 3:1 mixture of high glucose Dulbecco’s Modified Eagle’s Medium (DMEM) and Ham’s F-12 medium (Life Technologies) supplemented with 10% (*v*/*v*) fetal bovine serum (FBS, Atlanta Biologicals, Flowery Branch, GA, USA), 100 nM cholera toxin (Vibrio Cholerae, Type Inaba 569 B; Millipore, Burlington MA, USA), 5 μg/mL transferrin (Life Technologies), 0.4 μg/mL hydrocortisone (Sigma), 0.13 U/mL insulin (Sigma), 1.4 × 10^−4^ M adenine (Sigma), 2 × 10^−9^ M triiodo-L-thyronine thyronine (Sigma), 1× antibiotic-antimycotic (Life Technologies) and 10 ng/mL epidermal growth factor (EGF, BD Biosciences). Following 8–10 days of culture, the 3T3/J2 feeder layer was detached after a 10-min versine (Life Technologies) treatment leaving the KC colonies on the plate. Then the cells were treated with trypsin–EDTA (0.25%), followed by neutralization by a solution containing 10% FBS in PBS and plated in KC serum-free growth medium (KSFM, Epilife medium with Human Keratinocyte Growth Supplement; Life Technologies). The cells were further expanded up to 1–3 passages before induction to KC-NC.

#### 4.2.2. Induction of KC into KC-NC

KCs were seeded at a density of 8000–10,000 cells/cm^2^ on collagen type I coated dishes (10 μg collagen type I per cm^2^; BD Biosciences) in the presence of NC induction medium, comprising basal medium (EBM-2 medium; Lonza, Walkersville, MD, USA) plus 2% (*v*/*v*) FBS (Lonza), 10 μg per ml heparin (Lonza), 100 μg per ml ascorbic acid (Lonza), and 0.5 μg per ml hydrocortisone (Lonza), 1× Gentamicin/Amphotericin-B (Lonza) and supplemented with 10 ng/mL fibroblast growth factor 2 (FGF2; BD Biosciences) and 10 ng/mL insulin-like growth factor 1 (IGF1, Lonza). This media was named FI-media.

#### 4.2.3. Proliferation and Differentiation of KC-NC

After 7 days of induction treatment, the KC-NCs were selectively collected by digesting with warm (37 °C) Trypsin-EDTA (0.25%) for 1 min under the microscope, avoiding any KC colonies. The cells were neutralized by a solution containing 10% FBS in PBS and centrifuged at 300× *g* for 5 min. For the proliferation and bio-compatibility experiments, KC-NCs were cultured in FI media at 40 k/cm^2^ on the fibers. For differentiation towards SC lineage, KC-NCs were seeded at 20 k/cm^2^ in SC differentiation media consisting of EBM2 as basal medium, gentamycin (Lonza), 1% (*v*/*v*) FBS (Lonza), 50 ng/mL NRG1 (Sigma), 10 ng/mL BDNF (Life Technologies), 0.5× B27 Supplement (Life Technologies), 200 µg/mL ascorbic acid (Sigma), 0.5× glutamax (Life Technologies) for 14 days. KC-NC cells were derived from 3 donors to conduct the three experimental repeats.

#### 4.2.4. Immunocytochemistry

Cells were washed with cold PBS (4 °C) once before fixing with 4% (*v*/*v*) paraformaldehyde (10 min, room temperature; Sigma). For permeabilization, the samples were treated with 0.1% (*v*/*v*) Triton X-100 (Sigma) in PBS (10 min, room temperature). Next, the cells were blocked with 5% (*v*/*v*) goat serum (Life Technologies) in PBS and incubated overnight at 4 °C with the primary antibodies (Anti-GFAP, Catalog# AB5804, EMD Millipore, 1:400, rabbit polyclonal; Anti-myelin PLP or PLP1, Catalog# ab28486, Abcam, Cambridge, MA, USA, 1:200, rabbit; Anti-S100b, Catalog# ab52642, Abcam, 1:200, rabbit monoclonal; Anti-p75 NGFR, Catalog# ab245134, Abcam, 1:200, mouse monoclonal). The cells were washed three times before incubating with appropriate Alexa-Fluor conjugated secondary antibodies (Catalog #A11008, #A11031, 1:400 in 5% (*v*/*v*) goat serum, Life Technologies) for 1 h at room temperature. The cell nuclei were stained with Hoechst 33342 also known as Dapi (Catalog #H1399, 1:1000 in PBS, Life Technologies). Cells incubated with only secondary antibodies served as negative controls. Cells were stained with Alexa Fluor 488 Phalloidin also known as F-Actin (Catalog #A12379, 1:400 in 1% BSA, Life Technologies) to visualize the microfilaments of the cytoskeleton. After the permeabilization step (as described above), cells were blocked with 1% (*w*/*v*) BSA + 0.01% Triton X in PBS for 30 min, washed three times, and cell nuclei were stained with TO-PRO-3 Iodide (catalog# T3605, 1:1000 in PBS, Life Technologies) for 30 min.

#### 4.2.5. Fluorescence Microscopy and Image Analysis

Immunocytochemistry images were obtained using a Zeiss Axio Observer Z1 inverted microscope with a Hamamatsu ORCA-ER CCD camera. The images were captured using a fixed exposure time for each fluorescent dye for all the samples. Confocal images were collected by a white light laser scanning Leica Stellaris 5 (Leicamicrosystems, Wetzlar, Germany) with a 20× objective, and the z-stack depth was within 10 μm (step = 1 μm). Quantification of cell number and fluorescence intensity was measured using the ImageJ software.

#### 4.2.6. Statistical Analysis

The experiments were repeated three times independently with three different donors. The data are expressed as mean ± standard deviation (SD). GraphPad Prism^®^ software was used for the single-factor analysis of variance (ANOVA). * denotes *p* < 0.05, ** *p* < 0.005, *** *p* < 0.0005, **** *p* <0.0001 were considered statistically significant.

## 5. Conclusions

Taken together, our study shows enhanced cell attachment, alignment, and differentiation of epidermis-derived NC towards SC on nanofiber scaffolds. While electrospun nanofibers alone could promote alignment, SC differentiation required the presence of biological cues such as NRG1. Incorporation of biomechanical cues may further enhance the differentiation and induce processes such as axonal sorting, which are critical for myelination.

## Figures and Tables

**Figure 1 ijms-23-10834-f001:**
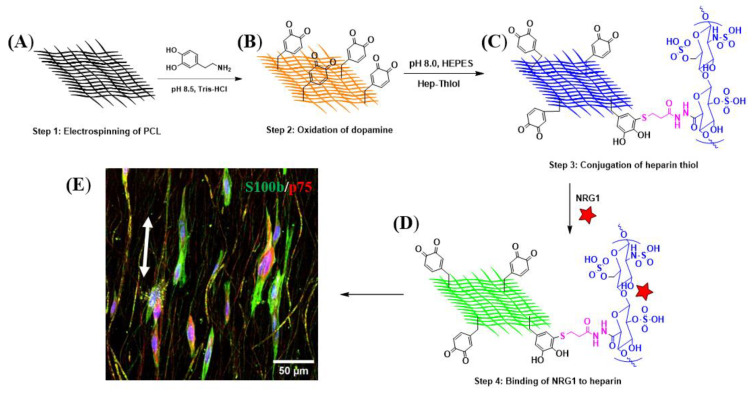
The schematic of designing a cell-instructive fiber for directing the KC-NC fate. (**A**) Electrospinning of PCL to obtain aligned fibers. (**B**) Preparation of PCL-PDA via oxidative polymerization of dopamine in the presence of PCL fibers; simplified structure of PDA is shown here. (**C**) Conjugation of Hep-thiol to PCL-PDA via Michael addition chemistry at physiological pH (7–8), (**D**) Immobilization of biological cues via their HBD. (**E**) Confocal microscopy image of the KC-NC grown on aligned NRG1 functionalized fibers and stained for S100b (green), p75 (red), and nuclei (blue). Double-arrow: Aligned fiber direction.

**Figure 2 ijms-23-10834-f002:**
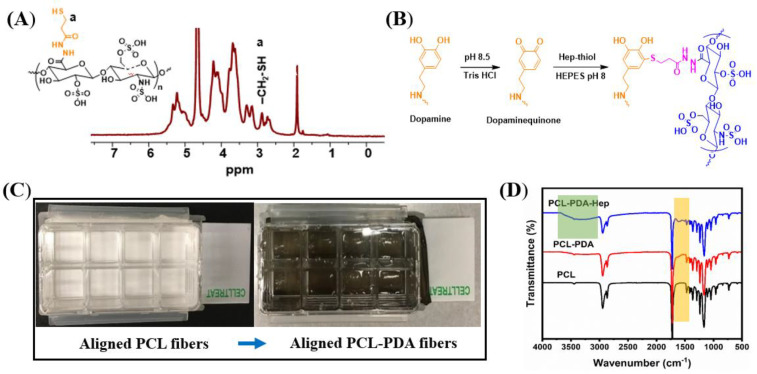
Surface modification and characterization of the aligned fibers. (**A**) ^1^H NMR analysis of Hep-thiol. (**B**) Chemical conjugation of Hep-thiol to PCL-PDA via Michael-addition chemistry. (**C**) PCL and PCL-PDA mounted in polystyrene chamber with movable glass base. (**D**) FTIR spectrum of PCL, PCL-PDA, and PCL-PDA-Hep.

**Figure 3 ijms-23-10834-f003:**
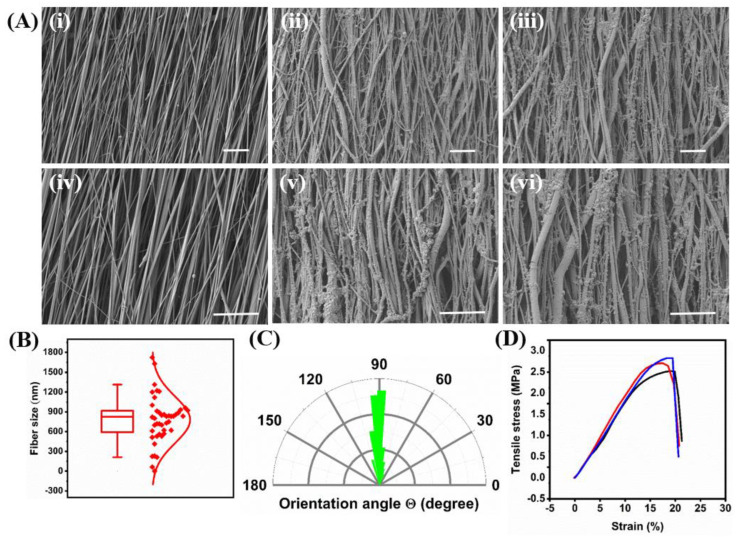
Fiber formation, size distribution, orientation, and mechanical strength. (**A**) SEM analysis of electrospun PCL (**i**,**iv**), PCL-PDA (**ii**,**v**), or PCL-PDA-Hep (**iii**,**vi**) fibers at two magnifications (top images were taken at 3000×; bottom images were taken at 5000×; scale bars = 10 µm). (**B**) Diameter distribution of PCL fibers. (**C**) The orientation angle of the PCL-PDA-Hep fibers was measured using image J software. (**D**) Stress-strain curve of PCL-PDA-Hep (n = 3 measurements are shown in red, blue, and black colors respectively).

**Figure 4 ijms-23-10834-f004:**
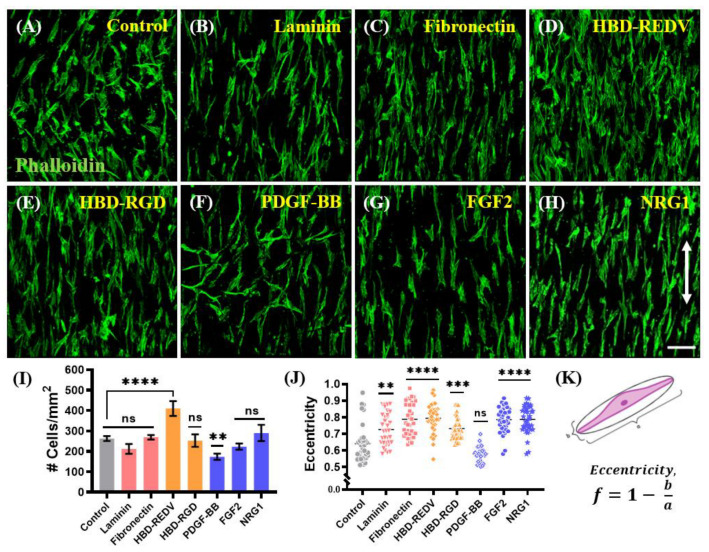
Effect of different cell substrates to guide KC-NC attachment and spreading on the fibers. (**A**–**H**) Confocal microscopy images of the KC-NC cells stained for their cytoskeletal microfilaments with actin phalloidin (green) on fibers with no coating (Control), two native ECM glycoproteins (laminin and fibronectin), two recombinantly designed bidomain fusion peptides (HBD-REDV and HBD-RGD), and three bioactive macromolecules (PDGF-BB, FGF2, and NRG1). Scale bar: 100 µm. (**I**) Quantification of the number of attached cells from n = 3 randomly selected fields per condition. Data are presented as mean ± SD. One-way analysis of variance (ANOVA) with Tukey post hoc test, n = 3 independent experiments. (**J**) Analysis of cell spreading quantified as eccentricity for each condition. Data are presented as mean ± SD of randomly selected cells. One-way analysis of variance (ANOVA) with Tukey post hoc test. n = 35 randomly selected cells from different fields, ns: *p* ≥ 0.05, **: *p* < 0.005, ***: *p* < 0.0005, ****: *p* < 0.0001). (**K**) Cell eccentricity, φ is calculated from measurements of the major and minor axis length using ImageJ. Double-arrow: direction of aligned fibers.

**Figure 5 ijms-23-10834-f005:**
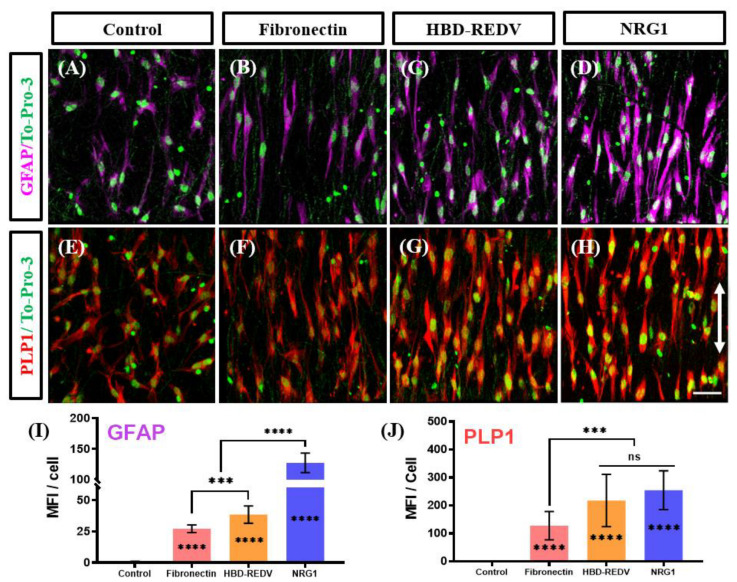
Evaluation of Schwann differentiation on the fibers conjugated with biological cues. Confocal microscopy images for the expression of astrocyte marker, glial fibrillary acidic protein, GFAP (**A**–**D**), and Schwann marker myelin proteolipid protein, PLP1 (**E**–**H**), on electrospun fibers with no coating (Control) (**A**,**E**), Fibronectin (**B**,**F**), HBD-REDV (**C**,**G**), and NRG1 (**D**,**H**), respectively. Scale bar = 100 µm. (**I**,**J**) Graph showing the quantification of the mean fluorescence intensity (MFI) for GFAP or myelin PLP1 per cell. Data are presented as Mean ± SD, One-way analysis of variance (ANOVA) with Tukey post hoc test. n = 20 randomly selected cells from three different fields. ns: *p* ≥ 0.05), ***: *p* < 0.0005), ****: *p* < 0.0001. Double-arrow: direction of aligned fibers.

**Figure 6 ijms-23-10834-f006:**
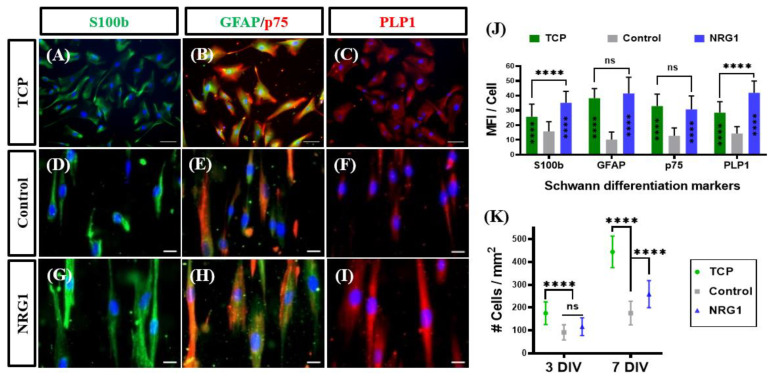
Effect of PCL immobilized NRG1 on KC-NC differentiation and proliferation toward towards Schwann cells. Immunofluorescence images of the expression of Schwann marker, S100b, GFAP, p75, and myelin PLP1 on (**A**–**C**) tissue culture plates (TCP), (**D**–**F**) electrospun fibers with no coating (Control), or (**G**–**I**) immobilized NRG1. (**A**–**C**) scale bars: 50 µm; (**D**–**I**) scale bars: 20 µm. (**J**) Quantification of the mean immunofluorescence intensity (MFI) of the indicated SC markers normalized per cell on TCP, control fibers, and NRG1 modified fibers. n = 50 randomly selected cells from different fields. (**K**) Quantification of average cell density at two-time points (3 DIV and & DIV, DIV = Day in-vitro) on TCP, control, or NRG1 modified fibers. Data are presented as Mean ± SD, One-way analysis of variance (ANOVA) with Tukey post hoc test. ns: *p* ≥ 0.05), ****: *p* <0.0001.

**Table 1 ijms-23-10834-t001:** Cloning primers for designing the recombinant bidomain fusion proteins.

Forward H2_BamHI	ATATGGATCCGCCATTCCTGCACCAACTGACCTGAAGTTCA
Reverse H2_HindIII	ATATAAGCTTATGTCCAATCAGGGGCTCGCTCTTCT
Forward Fragment-1 of HBD-RGD	TTGGACATAAGCTTGGCGGCGGAGGCAGTGGCCGTGGCGATTCCGGCGGCGGAGGCAGTGGCCGTGGCGATTCCGGCGGCGGAGGCAGTGGCCGTGG
Reverse Fragment-1 of HBD-RGD	TGGTGGTGCTCGAGATTATTAGGAATCGCCACGGCCACTGCCTCCGCCGCCGGAATCGCCACGGCCACTGCCTCCGCCGCCGGAATCG
Forward Fragment-2 of HBD-RGD	CGATTCCGGCGGCGGAGGCAGTGGCCGTGGCGATTCCGGCGGCGGAGGCAGTGGCCGTGGCGATTCCTAATAATCTCGAGCACCACCA
Reverse Fragment-2 of HBD-RGD	CCACGGCCACTGCCTCCGCCGCCGGAATCGCCACGGCCACTGCCTCCGCCGCCGGAATCGCCACGGCCACTGCCTCCGCCGCCAAGCTTATGTCCAA

## Data Availability

Not applicable.

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
