# Peer review of "Engineering Nanofiber Scaffolds with Biomimetic Cues for Differentiation of Skin-Derived Neural Crest-like Stem Cells to Schwann Cells"

_ijms, 2022, doi:10.3390/ijms231810834_

Round 1
Reviewer 1 Report
Dear Authors,
Manuscript ID: ijms-1790637, "Engineering nanofiber scaffolds with biomimetic cues for differentiation of skin-derived neural crest-like stem cells to Schwann cells," quite an interesting and important study. I acknowledge the authors' efforts and findings on PCL nanofibers coated with NGR1 that promote KC-NC differentiation to SC. On the other hand, PCL is a biocompatible polymer widely used in neural tissue engineering to support cell attachment and growth, promoting differentiation. My main concern is the advantages of coating PCL nanofibers with NRG1 over cell culture on PCL nanofibers in an NRG1-supplemented culture medium, which I believe the author should validate with additional experiments. The findings are well-narrated in the manuscript, but several limitations that, in my opinion, may and should be addressed to fully support the conclusions
General Comments
§ The text in the introduction and discussion is very lengthy. Please be concise.
§ Even though the authors claimed to have created nanofibers, Figure 3B shows that it is microfiber (about 800 mM). Keep the terms fiber, microfiber, or nanofiber consistent throughout the entire manuscript's results and discussion.
§ "Decoration or Decorated" words are too fancy. Instead, coating or layering makes good sense.
§ How long does this NRG1 remain immobilized on the PCL surface over the time in culture? Did you measure the release of NRG1 in culture medium over time? Is PCL immobilized with NRG1 appropriate for long-term culture?
Major Comments
§ I believe that the authors have used a human sample. Please provide an ethical approval code and include the letter as a supplementary file.
§ To claim the possible benefits of NRG1 coating in PCL for differentiation to SC, please include a comparison between PCL, PCL-PDA, and PCL-NRG1 nanofibers.
§ PCL Nanofibers do not appear to be homogeneous and smooth. Please include a high magnification image of the nanofibers for better visibility. Do you believe PDA and Hep-Thiol affect the surface of nanofibers? What was the thickness of the nanofiber mat and tensile property?
§ Did the authors neutralize the toxicity of HFIP solvents used to dissolve PCL? Please include a cell proliferation assay for primary SC cultured in PCL nanofibers alone and with PCL-PDA at different time points.
§ Include a morphological and characterization panel of keratinocytes utilizing immunostaining and PCR.
§ In figure 1E, please include data for cell attachment and growth comparison between PCL and NRG1 functionalized PCL.
§ Please include a high magnification image of Figure 2C for better visibility.
§ Please include FTIR analysis for NRG1 coated PCL fiber in Figure 2D.
§ In figure 3A, the descriptive panel below nanofibers is not visible. Please remove it and describe it in the figure legend. Also, please provide a magnified image of nanofibers after oxidization and coating instead of the same images in different magnifications.
§ Please include statistical data for orientation angle measurement using image J. How many deviations from right angles of nanofiber were analyzed to confirm proper alignment?
§ In figure 4, include labels starting from the IF image panel. Also, Include a high magnification image for each IF for better cell attachment visibility with fibers. Include raw statistical analysis data for cell quantification and eccentricity in the supplementary file.
§ In figure 5, include high magnification for better visibility. Also, provide raw data for 5B and 5C in supplementary files.
§ In figure 6, provide a high magnification image for control and NRG1 like in the TCP panel. Why is an expression of Schwann cell markers significantly low in PCL control compared to TCP? Does this mean nanofiber is not biocompatible to KC-NC differentiation to SC?
Best Wishes
Reviewer 2 Report
The manuscript described by Podder et al. reported PCL-based nanofiber scaffolds modified with dopamine, heparin and growth factors for the differentiation of keratinocyte-derived neural crest-like cells into Schwann cells. The authors observed cell attachment, alignment, and differentiation of keratinocyte-derived neural crest-like cells towards Schwann cells on the scaffolds, and the differentiation process was enhanced by NRG1 immobilization to the scaffolds.
Major points
1. The differentiation procedure described in this study should be compared with previous methods. Is there any difference in cell function?
2. NRG1 was effective for the differentiation of keratinocyte-derived neural crest-like cells into Schwann cells. In Fig. 6J, for comparison, NRG1 should be added to the medium for TCP and control (non-immobilized NRG1) cultures. Is immobilization of NRG1 on the scaffolds essential?
Minor points
“DI”, “Hep”, “NC” and “TCP” should be added to the abbreviation list.
P15, L6: “PDFG” => “PDGF”
Round 2
Reviewer 1 Report
Dear Authors,
The revised version makes sense to me, and I acknowledge the findings. However, I would appreciate the authors making a few minor changes to the final draft.
1. If its nanofiber, please correct "Fiber size (µm)" in the Y axis of Figure 3B to "Fiber size (nm)". Please make sure it's nanofiber before changing it.
2. In section 2.7, line 4, it should be "Fig. S4", not "Fig. S1". Please verify and correct it.
3. EDC/NHS has been used in the crosslinking of nanofibers as well as the structure of tissues. Please emphasize a few references in the results (section 4.1) or discussion. 1016/j.tice.2016.07.007, 1016/j.msec.2021.111938
Thank you
Author Response
Reviewer-1
Dear Authors,
The revised version makes sense to me, and I acknowledge the findings. However, I would appreciate the authors making a few minor changes to the final draft.
- If it’s nanofiber, please correct "Fiber size (µm)" in the Y axis of Figure 3B to "Fiber size (nm)". Please make sure it's nanofiber before changing it.
Ans: Corrected. Thank you for pointing this out.
- In section 2.7, line 4, it should be "Fig. S4", not "Fig. S1". Please verify and correct it.
Ans: Corrected. Thank you for pointing this out.
- EDC/NHS has been used in the crosslinking of nanofibers as well as the structure of tissues. Please emphasize a few references in the results (section 4.1) or discussion. 1016/j.tice.2016.07.007, 1016/j.msec.2021.111938
Ans: Thank you for the feedback. There were no suitable paragraphs in the discussion or results sections that could accommodate the necessary references supporting EDC/NHS crosslinking of fibers. So, we have add references related to crosslinking, including EDC-NHS in the Introduction (2nd paragraph, lines #3,4) of the paper (page #3).
Reviewer 2 Report
The manuscript has been revised properly.
Author Response
Reviewer-2
The manuscript has been revised properly.